# Verification of a Novel Approach to Predicting Effects of Antibiotic Combinations: In Vitro Dynamic Model Study with Daptomycin and Gentamicin against *Staphylococcus aureus*

**DOI:** 10.3390/antibiotics9090538

**Published:** 2020-08-25

**Authors:** Maria V. Golikova, Elena N. Strukova, Yury A. Portnoy, Stephen H. Zinner, Alexander A. Firsov

**Affiliations:** 1Department of Pharmacokinetics & Pharmacodynamics, Gause Institute of New Antibiotics, 11 Bolshaya Pirogovskaya Street, 119021 Moscow, Russia; lena-stru@inbox.ru (E.N.S.); yaportnoy@gmail.com (Y.A.P.); kindyn@gmail.com (A.A.F.); 2Department of Medicine, Harvard Medical School and Mount Auburn Hospital, 330 Mount Auburn St., Cambridge, MA 02138, USA; szinner@mah.harvard.edu

**Keywords:** daptomycin–gentamicin combination, in vitro model, anti-staphylococcal effect

## Abstract

To explore whether susceptibility testing with antibiotic combinations at pharmacokinetically derived concentration ratios is predictive of the antimicrobial effect, a *Staphylococcus aureus* strain was exposed to daptomycin and gentamicin alone or in combination in multiple dosing experiments. The susceptibility of the *S. aureus* strain to daptomycin and gentamicin in combination was tested at concentration ratios equal to the ratios of 24 h areas under the concentration–time curve (AUC_24_s) of antibiotics simulated in an in vitro dynamic model in five-day treatments. The MICs of daptomycin and gentamicin decreased in the presence of each other; this led to an increase in the antibiotic AUC_24_/MIC ratios and the antibacterial effects. Effects of single and combined treatments were plotted against the AUC_24_/MIC ratios of daptomycin or gentamicin, and a significant sigmoid relationship was obtained. Similarly, when the effects of single and combined treatments were related to the total exposure of both drugs (the sum of AUC_24_/MIC ratios (∑AUC_24_/MIC)), a significant sigmoid relationship was obtained. These findings suggest that (1) the effects of antibiotic combinations can be predicted by AUC_24_/MICs using MICs of each antibacterial determined at pharmacokinetically derived concentration ratios; (2) ∑AUC_24_/MIC is a reliable predictor of the antibacterial effects of antibiotic combinations.

## 1. Introduction

The extensive use of daptomycin in clinical practice has led to decreased daptomycin effectiveness due to the emergence of bacterial resistance [1,2,3,4,5,6,7,8,9,10,11,12,13,14,15]. Daptomycin treatment failures have been reported in patients with *Staphylococcus aureus* infections after prolonged daptomycin exposure [1,2,3,4,5,6,7,8] and also following vancomycin administration that was associated with vancomycin-resistant *S. aureus* [9,10,11,12,13,14,15]. Similarly, infections caused by vancomycin-resistant enterococci with lowered susceptibility to daptomycin can be difficult to treat even at high dosing regimens of daptomycin [16]. Combination antibiotic therapy can be a valuable option to improve the antibacterial effectiveness of daptomycin. In our previous study, the combination of daptomycin with rifampin used at subtherapeutic doses was characterized by an increased anti-staphylococcal effect and the pharmacodynamic interaction was interpreted as synergistic [17]. The enhanced efficiency of daptomycin–rifampicin combinations was predicted by the ratio of the 24-area under the concentration–time curve (AUC_24_) to the MIC of daptomycin determined in the presence of rifampin. Susceptibility testing of *S. aureus* to daptomycin or rifampin in combination was performed at pharmacokinetically derived antibiotic concentration ratios, which were strongly related to the daptomycin-to-rifampicin AUC_24_ ratios used in subsequent pharmacokinetic simulations. 

Verification of this approach with daptomycin in combination with other anti-staphylococcal antibiotics such as gentamicin is of interest. The synergistic interaction between daptomycin (as a cell membrane targeting antibacterial effects) and gentamicin (ribosomal targeting) can be hypothesized based on existing knowledge that daptomycin acts on bacterial cell membrane and increases its envelope permeability to another drug such as gentamicin (uptake effect) [18]. However, some previously conducted studies exploring the synergistic potential of daptomycin–gentamicin combinations [19,20,21,22,23,24,25,26,27] yielded controversial results. In some checkerboard studies with staphylococci, daptomycin plus gentamicin combinations were mostly synergistic [19], while in others, they were additive or indifferent [20,21]. In static time–kill experiments with these drugs, synergy was reported as prevalent [22,23]. Similar discrepancies were observed in studies using in vitro dynamic models that consider differences in daptomycin and gentamicin pharmacokinetics [24,25,26,27]. The effects of therapeutic exposures of daptomycin–gentamicin combinations were either enhanced or improved [24,25,26] but were antagonistic in one study [27]. Moreover, the in vivo anti-staphylococcal efficiency of daptomycin was not enhanced by the addition of gentamicin during treatment of experimental endocarditis in humanized rabbits [23].

To explore whether susceptibility testing with antibiotic combinations at pharmacokinetically derived concentration ratios is predictive of the antimicrobial effect, a clinical *S. aureus* strain was exposed to daptomycin and gentamicin in an in vitro dynamic model that simulates single and combined treatments. As therapeutic daptomycin exposures were characterized by a pronounced effect against staphylococci [28], subtherapeutic doses, which provide low antibacterial activity, were used in the present study. In contrast, gentamicin pharmacokinetics were simulated at therapeutic exposures, given previous in vitro studies that reported low [25,26] or moderate [29,30] anti-staphylococcal efficacy even at high peak concentrations at extended dosing interval regimens.

## 2. Results

### 2.1. MICs of Daptomycin and Gentamicin Alone and in Combination

Daptomycin-to-gentamicin concentration ratios used in MIC determinations corresponded to AUC_24_ ratios used in the pharmacodynamic simulations. The MICs of daptomycin and gentamicin alone and in combination are listed in Table 1. As follows from the table, the MIC of daptomycin in the presence of gentamicin decreased 4- to 16-fold when the aminoglycoside portion in the combination prevailed (daptomycin-to-gentamicin concentration ratios of 1:1.5, 1:2 and 1:5). When the daptomycin portion in the combination prevailed (1.5:1), daptomycin MIC decreased only two-fold compared to the MIC determined in the absence of gentamicin. At the same time, gentamicin MICs under the influence of daptomycin decreased 1.5- to 3-fold depending on the antibiotic concentration ratios. Changes in *S. aureus* susceptibility to daptomycin and gentamicin in the presence of each other were reflected in fractional inhibitory concentration (*FIC*) analysis (Table 1). As seen in the table, with increasing gentamicin portions in the combination from 0.67 (ratio 1.5:1) to 1.5 (ratio 1:1.5), the total (summation) fractional inhibitory concentration index (∑*FIC)* fell from 1.18 to 0.48, revealing a change in the type of antibiotic interaction from indifferent to synergistic. Subsequent increases in the relative portion of gentamicin in the combination led to systematic increases of ∑*FIC* index from 0.48 to 0.64 and from 0.64 to 0.70 at concentration ratios 1:2 and 1:5, respectively, reversing to an indifferent interaction. Thus, the lower ∑*FIC* showing synergy was associated with the proportion of antibiotics in the combination with relatively higher gentamicin concentrations.

### 2.2. Antibiotic Pharmacodynamics with S. aureus

Time–kill kinetics of *S. aureus* 293 exposed to daptomycin and gentamicin alone or in combination in the in vitro dynamic model are shown in Figure 1. As seen in the figure, single daptomycin treatments at AUC_24_ 30 or 100 mg·h/L were associated with only a slight anti-staphylococcal effect within the first 12 or 24 h from the beginning of the experiment, respectively, followed by regrowth to cell density that exceeded the initial numbers. Gentamicin produced more pronounced anti-staphylococcal effects than daptomycin and within the first 24 h of treatment reduced initial cell numbers to either 4 log CFU/mL (AUC_24_ 65 mg·h/L) or below the limit of detection (AUC_24_ 160 mg·h/L). However, rapid initial killing by gentamicin was followed by fast regrowth and the numbers of surviving organisms at the end of the treatment were equal or close to the initial inocula (8 log CFU/mL at AUC_24_ 65 mg·h/L or 7 log CFU/mL at AUC_24_ 160 mg·h/L). However, all combined treatments with daptomycin and gentamicin were associated with more pronounced antibacterial effects than respective mono-treatments. Comparing the numbers of surviving staphylococci in treatments with the lower dose of gentamicin alone (as the more active single agent) and in combined treatments (regimens G65 versus D30+G65 and D100+G65), independent of the relative amount of daptomycin, the antibiotic combinations produced more pronounced initial bacterial killing (reduction to the limit of detection) and weaker regrowth (Figure 1a,c). At the higher gentamicin AUC_24_ (G160), the initial killing of *S. aureus* was the same in mono- and in combined treatments; cell numbers reached the limit of detection during the first 24 h. However, the regrowth phase was delayed in combined treatments and at the end of the experiment cell density reached 5.5 log CFU/mL (ratio 1:5, regimen D30+G160) versus 7 log CFU/mL with gentamicin alone (G160); bacterial regrowth was fully suppressed with regimen D100+G160 (ratio 1:1.5) (Figure 1b,d).

### 2.3. Pharmacodynamic Bliss Independence-Based Drug Interaction Analysis

To accurately assess the advantages of combined therapy, Bliss independence-based drug interaction analysis was performed using data obtained in the pharmacodynamic experiments (Figure 2). In all combination regimens, *E*_DG_ was significantly higher than *E*_IND_ and the respective lower limits of 95% CI always were positive. Depending on the combination regimen, ∆*E* varied from 8 to 22%, with the maximum ∆*E* value in regimen D100+G160 (1:1.5 AUC_24_ ratio).

### 2.4. ABBC–AUC_24_/MIC Relationships

The enhancement of anti-staphylococcal efficiency of daptomycin and gentamicin combinations could be explained by increased *S. aureus* susceptibility to the antibacterials in the presence of each other (Table 1). In terms of AUC_24_/MIC, the actual exposures of daptomycin given with gentamicin and gentamicin given with daptomycin are much greater than in single treatments. For example, daptomycin AUC_24_/MICs under the influence of gentamicin increased from 60 h (30/0.5, regimen D30) to 500 h (30/0.06 h, regimen D30+G65) and 1000 h (30/0.03 h, regimen D30+G160). Similarly, gentamicin AUC_24_/MICs under the influence of daptomycin increased from 260 h (65/0.25, regimen G65) to 380 h (65/0.17, regimen D100+G65) and 500 h (65/0.13, regimen D30+G65). Similar AUC_24_/MIC augmentations were obtained with other daptomycin plus gentamicin combinations (see Table 1). 

To explore if the increase in the area between the control growth curve and each time–kill curve (ABBC) observed with the antibiotic combinations can be explained by these greater AUC_24_/MIC ratios, ABBCs were plotted against AUC_24_/MICs simulated in both single and combined antibiotic treatments (Figure 3). As seen in the figure, the AUC_24_/MICs of daptomycin (Figure 3a) and gentamicin (Figure 3b) used in combination were shifted to the right along the abscissa, i.e., to the area of higher AUC_24_/MIC values providing reasonable sigmoid relationships with ABBC.

## 3. Discussion

Daptomycin plus gentamicin combinations were characterized primarily as indifferent by the results of *FIC* analyses (∑*FIC*s 0.64–1.18); at only one concentration ratio with slight prevalence of gentamicin (1:1.5) did the antibiotic combination demonstrate enhanced anti-staphylococcal activity considered as synergistic (∑*FIC* 0.48) (Table 1). More encouraging conclusions were drawn from 5-day dynamic model simulations where the anti-staphylococcal effects observed in each combined therapy (expressed as % of the ABBC responsible for the maximal antibacterial effect) was greater than expected assuming the Bliss independence principle (∆*E* from 8 to 22%), and statistically significant synergism was confirmed (Figure 2). The enhancement of antibacterial effects in combination therapy was more pronounced with the daptomycin-to-gentamicin AUC_24_ ratio 1:1.5 (regimen D100+G160) compared to other combined treatment regimens. In addition, only at regimen D100+G160 was bacterial regrowth fully prevented. Interestingly, the same antibiotic concentration ratio (1:1.5) was assessed as synergistic by the *FIC* analysis. In addition, the relatively higher ∑*FIC* indices for other antibiotic combinations that predicted only indifferent interactions (0.64–1.18) were consistent with the relatively smaller ∆*E*s determined for respective combined treatments (8–13%) as compared with ∆*E* at AUC_24_ ratio 1:1.5 (22%).

The results of *FIC* analyses are consistent with the pharmacodynamic data with respect to daptomycin and gentamicin interactions when the antibiotics were used in combination. It is highly likely that this observation is due to the use of the same antibiotic concentration ratios in both susceptibility testing and AUC_24_ ratios in the pharmacokinetic simulations. The predictive potential of susceptibility testing determined at pharmacokinetically derived concentration ratios was confirmed in our previous study with daptomycin and rifampin [17] where the ∑*FIC* indices adequately reflected the ability of pharmacodynamic data to predict synergism or additivity.

Discrepancies between the results of synergy studies reported by other investigators with various antibiotic combinations including daptomycin plus gentamicin have been discussed [31,32,33]. These discrepancies were documented even when checkerboard and time–kill assays were performed in the same laboratory [20,34,35,36]. Most likely, this may have resulted from the use of arbitrarily chosen antibiotic concentration ratios that usually do not match each other in different methods. Results obtained using different ratios for checkerboard and time–kill assays may differ significantly. In light of our current data, the most effective concentration ratios obtained in checkerboard studies should be used in time–kill assays (including in dynamic models). From this viewpoint, when the results of these two methodologies with various daptomycin containing combinations including daptomycin plus gentamicin were compared against *Staphylococci* and *Enterococci* [20], consistent predicted interactions were achieved only at equal antibiotic concentration ratios.

Enhancement or improvement in daptomycin efficacy against daptomycin-susceptible *S. aureus* strains in the presence of gentamicin was described previously in studies with in vitro dynamic models [24,25,26,27]. However, advantages of combined therapy compared to mono-treatments were seen only in the first 24–36 h of observation as continued daptomycin alone provided sustained bactericidal activity up to the 72 h [24] or 96 h endpoint [25,26]. This is due most likely to the high therapeutic AUC_24_s of daptomycin (750–1750 mg·h/L) that were responsible for the pronounced antibacterial effect. As a result, the relative portion of daptomycin combined with gentamicin in terms of the AUC_24_ ratio was quite prominent: daptomycin-to-gentamicin AUC_24_ ratios of 15–30:1. Comparative efficacy between regimens with and without gentamicin could not be distinguished over the duration of the experiment, making it difficult to adequately assess the actual synergistic potential of the combination regimens. The relative deficiency of gentamicin in the daptomycin combinations probably played a key role in the failure of the combination to improve daptomycin’s weak efficacy against daptomycin-non-susceptible *S. aureus* strains [26]. The same considerations could apply to the results of an in vivo study in humanized rabbits where the addition of gentamicin did not enhance daptomycin efficacy due to a disproportional daptomycin-to-gentamicin AUC_24_ ratio (850 to 15 mg·h/L, i.e., 60-fold difference) [23].

In the current study, the synergistic potential of daptomycin–gentamicin combinations was observed with the results of both checkerboard and pharmacodynamic experiments at comparable antibiotic concentrations or AUC_24_s, particularly with the use of subtherapeutic daptomycin exposures. Perhaps a way to improve the efficacy of daptomycin–gentamicin combinations is to use the antibacterials at doses that provide comparable AUC_24_s, usually representing lower daptomycin AUC_24_s. However, our results should be considered provisional and need additional confirmation in other experimental models. Of note, the *S. aureus* strains used in an experimental endocarditis model produced biofilms that could influence the pharmacodynamic interactions of daptomycin and gentamicin; this might have been responsible for the antagonism observed in that study [27].

In the current study, the potentiation of anti-staphylococcal effects of daptomycin–gentamicin combinations relative to monotherapy with the respective drugs was consistent with increased *S. aureus* susceptibility to these antibiotics in the presence of each other. This observation was confirmed by relating antibacterial effects of the combinations with respective antibiotic AUC_24_/MICs calculated by using MICs determined at pharmacokinetically derived concentration ratios (Figure 3). As seen in the figure, ABBCs determined in combined and single treatments could be plotted on the same sigmoid graph against AUC_24_/MIC of daptomycin (Figure 3a) or gentamicin (Figure 3b) with *r*^2^ 0.907 or 0.730, respectively. As a result, reasonable AUC_24_/MIC–ABBC relationships were demonstrated: the higher daptomycin or gentamicin AUC_24_/MIC, the greater the bacterial killing.

Usual predictions of antibacterial effects of daptomycin–gentamicin combinations are based on AUC_24_/MIC-dependent relationships for each drug separately and do not consider that the observed effect is provided by both drugs simultaneously. To more appropriately relate the antibacterial effect with total exposure of both drugs, the sum of simulated daptomycin and gentamicin AUC_24_/MIC ratios in combined treatments was calculated and noted as ∑AUC_24_/MIC. Using pooled data, a sigmoid relationship between ∑AUC_24_/MIC and ABBC was obtained (Figure 4). Despite finding that gentamicin alone produced a relatively low AUC_24_/MIC–ABBC relationship (*r*^2^ 0.730) compared to daptomycin (*r*^2^ 0.907; Figure 3), the ∑AUC_24_/MIC–ABBC relationship for the combined data was described with a high correlation coefficient squared (*r*^2^ 0.915; Figure 4). Using a single relationship for the combined drugs is more appropriate since it considers cumulative exposures responsible for the total antibacterial effect.

The use of only one staphylococcal strain is a limitation of this study. Additional experiments are needed with more *S. aureus* strains exposed to daptomycin–gentamicin combinations as studied here. Along with that, relatively high limit of accurate detection of living bacteria in the system (2 × 10^3^ CFU/mL) throughout the observation period is another limitation of this study, as we could not see differences between the antibacterial effects of dosing regimens below this threshold. This could be important in determining the true antibacterial potential of the antibiotics (alone or in combination). Another limitation is that we did not consider any effect of protein binding (PB), which might influence antibacterial activity of the drugs. Unfortunately, the optimal method for properly correcting simulated profiles for PB is still undefined. Our detailed explanation of why we do not adjust the simulated antibiotic AUC/MICs for PB is described in a previous study with daptomycin [28].

## 4. Materials and Methods 

### 4.1. Antimicrobial Agents and Bacterial Strain

Daptomycin powder was purchased from Acros Organics (Geel, Belgium); gentamicin sulfate was purchased from PhytoTechnology Laboratories (Shawnee Mission, KS, USA). Clinical isolate *S. aureus* 293 was used in the study. 

### 4.2. Antibiotic Dosing Regimens and Simulated Pharmacokinetic Profiles

Simulated AUC_24_s of daptomycin both in single and combined treatments corresponded to sub-therapeutic dosing regimens and were equal to 30 and 100 mg·h/L (single treatments were designated as D30 and D100, respectively). Gentamicin AUC_24_s in mono and combined treatments corresponded to once daily doses 5 mg/kg and 7 mg/kg. The respective AUC_24_s, 65 and 160 μg·h/mL, used in pharmacokinetic simulations (single treatments were designated as G65 and G160, respectively), were calculated using peak serum gentamicin concentrations reported in human studies (16.6 and 39.8 μg/mL, respectively, [37,38]) and t_1/2_ (3 h, [38]). The simulated combined treatments included: daptomycin AUC_24_ 30 mg·h/L + gentamicin AUC_24_ 65 or 160 mg·h/L (regimens D30+G65 and D30+G160, respectively), daptomycin AUC_24_ 100 mg·h/L + gentamicin AUC_24_ 65 or 160 mg·h/L (regimens D100+G65 and D100+G160).

With all dosing regimens, a series of monoexponential profiles that mimic once-daily dosing of daptomycin with a half-life of 9 h [39] and gentamicin with a half-life of 3 h alone or in combination were simulated for 5 consecutive days.

### 4.3. In Vitro Dynamic Model

A previously described dynamic model was used in simulations of single drug treatments with daptomycin and gentamicin [40]. Briefly, the model consists of two connected flasks, one containing fresh Mueller–Hinton broth supplemented with 50 mg of Ca^2+^/L (CSMHB), because daptomycin antimicrobial activity is influenced by the presence of Ca^2+^ [41], and the other with a magnetic stirrer, the central unit, with the same broth containing either a bacterial culture alone (growth control experiment) or a bacterial culture plus antibiotic (killing/regrowth experiments). Peristaltic pumps circulated fresh nutrient medium to and from the central unit (volume 100 mL) at a flow rate of 7.7 mL/h for daptomycin or 23.2 mL/h for gentamicin.

To simulate combination treatments, the model was modified according to the Blaser and Zinner principle to provide simultaneous mono-exponential elimination of daptomycin and gentamicin [42]. The model was supplemented with an additional 200 mL flask with fresh CSMHB containing daptomycin at initial concentrations equal to those in the central unit. Peristaltic pumps circulated fresh nutrient medium to and antibiotic-containing medium (both daptomycin and gentamicin) from the central unit at a flow rate of 23.2 mL/h that corresponds to the antibiotic with the shorter half-life, i.e., gentamicin. To compensate for a too rapid decrease in concentrations of antibiotic with the longer half-life (daptomycin), peristaltic pumps circulated fresh medium to and daptomycin containing medium from the additional flask to the central unit at a flow rate of 15.5 mL/h (23.2 mL/h–7.7 mL/h).

The operation procedure used in the pharmacodynamic experiments was as described elsewhere [43]. Each experiment was performed at least in duplicate. Antibiotic dosing and sampling of the central unit were processed automatically, using computer-assisted controls. The system was filled with sterile CSMHB and placed in an incubator at 37 °C. The central unit was inoculated with an 18 h culture of *S. aureus* 293. After a 2 h incubation, the resulting exponentially growing cultures reached ~10^8^ colony-forming units (CFU)/mL (10^10^ CFU per 100 mL central unit). Then, antibiotics were administered into the central unit of the model. The duration of each experiment was 120 h.

### 4.4. Susceptibility Testing

Susceptibility testing was performed in triplicate using broth microdilution techniques according to Clinical and Laboratory Standards Institute (CLSI) methods [44]. After 24 h post-exposure with the organism grown in CSMHB, trays with twofold dilutions with known daptomycin and/or gentamicin concentrations were inoculated with *S. aureus* for a final concentration of approximately 5 × 10^5^ CFU/mL. The plates were incubated at 37 °C for 18 h.

With each antibiotic combination, stock solution concentration ratios of daptomycin to gentamicin corresponded to their AUC_24_ ratios used in the pharmacokinetic simulations and were as follows: 1.5:1 (regimen D100+G65), 1:1.5 (regimen D100+G160), 1:2 (regimen D30+G65) and 1:5 (D30+G160). The same concentration ratios were maintained in each subsequent dilution.

To determine the type of interaction between daptomycin and gentamicin in combination, the *FIC* for each antibiotic in each combination and Σ*FIC* were determined as:*FIC* of daptomycin (*FIC_D_*) = MIC of daptomycin in combination/MIC of daptomycin alone
*FIC* of gentamicin (*FIC_G_*) = MIC of gentamicin in combination/MIC of gentamicin alone
Σ*FIC* = *FIC_D_* + *FIC_G_*

Synergism was defined as a Σ*FIC* index ≤0.5, indifference was defined as a Σ*FIC* index >0.5 but ≤4.0, and antagonism was defined as a Σ*FIC* index >4.0 [45].

### 4.5. Quantitation of the Antimicrobial Effect and Its Relationships with AUC_24_/MIC Ratios

In each experiment, bacteria-containing medium from the central unit of the model was sampled to determine bacterial concentrations throughout the observation period. Samples (100 µL) were serially diluted as appropriate and 100 µL was plated onto Mueller–Hinton agar plates, which were placed in an incubator at 37 °C for 24 h. The lower limit of accurate detection was 2 × 10^3^ CFU/mL (equivalent to 20 colonies per plate).

Based on time–kill data, the area between the control growth curve and each time–kill curve (ABBC) [46] was determined from the beginning of treatment to 120 h.

Daptomycin or gentamicin AUC_24_/MIC relationships with ABBC observed in single and combined antibiotic treatments (merged data) were fitted by the sigmoid function:*Y* = *Y*_0_ + *a*/{1+exp [−(*x* − *x*_0_)/*b*]}(1)
where *Y* is ABBC, *x* is log (AUC_24_/MIC); *Y*_0_ and *a* are the minimal and maximal values of the ABBC, respectively; *x*_0_ is *x* corresponding to *a*/2; and *b* is a parameter reflecting sigmoidicity. 

All calculations were performed using SigmaPlot 12 software.

### 4.6. Pharmacodynamic Bliss Independence-Based Drug Interaction Analysis 

The Bliss independence principle was used to analyze the daptomycin–gentamicin interaction, assuming that the drugs do not interact with each other when used in combination [47,48,49]. Bliss independence is described by the equation:*E*_IND_ = *E*_D_ + *E*_G_ − *E*_D_ × *E*_G_(2)
where *E*_IND_ is the expected anti-staphylococcal effect of daptomycin–gentamicin combinations calculated using the effects of respective mono-treatments with daptomycin (*E*_D_) and gentamicin (*E*_G_).

The *E*_D_, *E*_G_ and antibacterial effects observed in combined treatments with daptomycin plus gentamicin (*E*_DG_) were calculated by dividing the respective experimental ABBCs to the ABBC responsive for maximal antibacterial effect (the area between the control growth curve and the line drawn at the lower limit of detection determined from 0 to 120 h) and expressed in %. The difference between the *E*_DG_ and *E*_IND_ (∆*E*) reflects the type of interaction between the daptomycin and gentamicin in each combination and is considered to be as follows: (i) Bliss synergy when ∆*E* and the lower limit of the 95% confidence interval (CI) >0, (ii) Bliss antagonism when ∆*E* and the upper limit of the 95% CI <0, (iii) Bliss independence in any other case when the 95% CI of ∆*E* includes 0.

## 5. Conclusions

This study suggests that (1) combinations of daptomycin with gentamicin against *S. aureus* were synergistic according to Bliss independence-based analysis of pharmacodynamic experiments and also by the results of *FIC* analyses; (2) the enhanced antibacterial effects of the combinations relative to mono-treatments are likely due to the increased susceptibility of *S. aureus* both to daptomycin and gentamicin in the presence of each other; (3) the antibacterial effects of antibiotic combinations can be predicted by AUC_24_/MICs calculated with the use of MICs of each antibiotic determined at pharmacokinetically derived concentration ratios; and (4) ∑AUC_24_/MIC is a reliable predictor of antibacterial efficacy of antibiotic combinations as it considers the summed exposures of both drugs responsible for the total antibacterial effect.

## Figures and Tables

**Figure 1 antibiotics-09-00538-f001:**
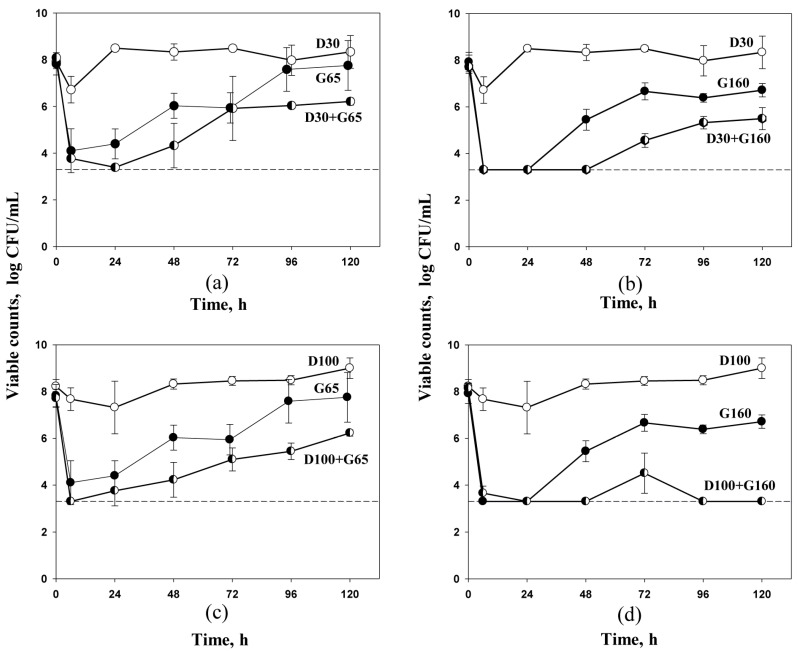
Time–kill curves of daptomycin and gentamicin, alone and in combination against *S. aureus* 293. Dosing regimens are indicated at each curve. Dotted lines indicate the limit of detection. Data are presented as arithmetic means ± standard deviations. (**a**)—regimens D30, G65 and D30+G65; (**b**)—regimens D30, G160 and D30+G160; (**c**)—regimens D100, G65 and D100+G65; (**d**)—regimens D100, G 160 and D100+G160.

**Figure 2 antibiotics-09-00538-f002:**
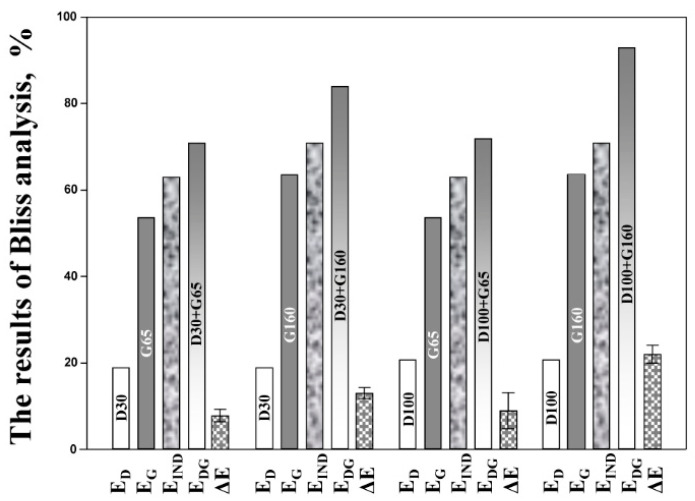
The results of Bliss independence drug interaction analysis of daptomycin and gentamicin combinations in an in vitro dynamic model.

**Figure 3 antibiotics-09-00538-f003:**
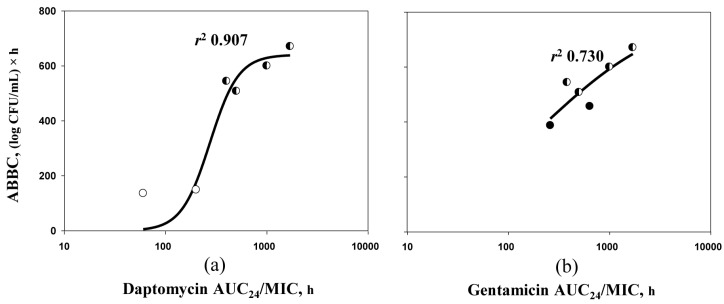
AUC_24_/MIC-related areas between the control growth curve and each time–kill curve (ABBCs) observed with *S. aureus* 293 in single treatments with daptomycin (open circles) or gentamicin (filled circles) and in combined treatments (half-filled circles). The relationships fit by Equation (1): (**a**) *Y*_0_ = 0, *x*_0_ = 2.438, *a* = 640.7, *b* = 0.1409 and (**b**) *Y*_0_ = 0, *x*_0_ = 2.385, *a* = 804.0, *b* = 0.5956.

**Figure 4 antibiotics-09-00538-f004:**
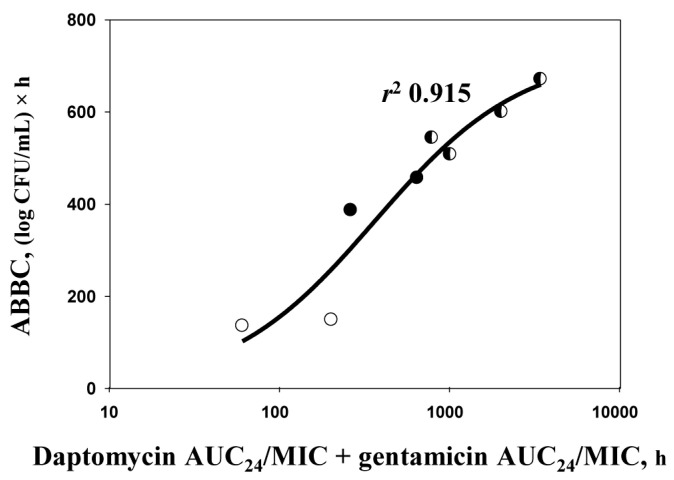
The sum of AUC_24_/MICs of daptomycin and gentamicin related to the ABBCs observed with *S. aureus* 293 in single treatments with daptomycin (open circles) or gentamicin (filled circles) and in combined treatments (half-filled circles). The relationships fit by Equation (1): *Y*_0_ = 0, *x*_0_ = 2.559, *a* = 728.9, *b* = 0.4316.

**Table 1 antibiotics-09-00538-t001:** MICs of daptomycin and gentamicin alone or in combination against *Staphylococcus aureus* 293 and respective *FIC* indices.

Dosing Regimen	Daptomycin-to-Gentamicin AUC24 Ratio	MIC (mg/L)		*FIC*		
		Daptomycin	Gentamicin	*FIC* _D_	*FIC* _G_	Ʃ*FIC*
D30	-	0.5	-	-	-	-
D100	-		-	-	-	-
G65	-	-	0.25	-	-	-
G160	-	-		-	-	-
D100 + G65	1.5:1	0.25	0.17	0.5	0.68	1.18
D100 + G160	1:1.5	0.06	0.09	0.12	0.36	0.48
D30 + G65	1:2	0.06	0.13	0.12	0.52	0.64
D30 + G160	1:5	0.03	0.16	0.06	0.64	0.70

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
