# Peer review of "Verification of a Novel Approach to Predicting Effects of Antibiotic Combinations: In Vitro Dynamic Model Study with Daptomycin and Gentamicin against *Staphylococcus aureus"

_antibiotics, 2020, doi:10.3390/antibiotics9090538_

Round 1
Reviewer 1 Report
This study gave the combination of daptomycin and gentamycin combinations against the styphylococcal. This study is sound and can be accepted after addressing the following comments;
- comparison with earlier reports necessary
- Title is too lengthy, revise if necessary
- Figure 1 x axis and y axis need to be gave separately to each figure.
- Why the combination is working better than individual drugs
- More detailed conclusions are need to be given
- 5. Figure 2 x axis very difficult to see, modify the figure
Author Response
Response to Reviewer 1 Comments
Point 1: Comparison with earlier reports necessary.
Response 1: Of note, in the introduction and discussion sections we analyzed in detail existing papers aimed at assessing efficacy of daptomycin-gentamicin combinations in vitro and in vivo in experiments with infected animals (Lines 53-63, 177-203, 209-212). We believe that the actual data reported in the cited studies are contradictory, since in checkerboard and in static and dynamic time-kill experiments (mentioned in Introduction and Discussion sections) the entire spectrum of interactions between daptomycin and gentamicin were reported, from synergism to antagonism. This did not allow us either to adequately assess the actual synergistic potential of daptomycin and gentamicin combinations or to compare these data with our results. Moreover, because of the principal role of antibiotic concentration ratios in determining the antibacterial effect of combinations (this sentence is considered in Discussion section, Lines 170-187), direct comparison of the results of synergy testing or pharmacodynamic experiments reported by other investigators and those presented in the current research would not be appropriate. This is primarily because we do not have any information about antibiotic concentration ratios used in most of these studies. Where these ratios were provided, they appeared quite different from those used in the current research (Lines 192-203).
Point 2: Title is too lengthy, revise if necessary.
Response 2: We agree with Reviewer’s comment. We suggest replacing the current title with the following: “Verification of a novel approach to predicting the effects of antibiotic combinations: in vitro dynamic model study with daptomycin and gentamicin against Staphylococcus aureus.”
Point 3: Figure 1 x-axis and y-axis need to be gave separately to each figure.
Response 3: We agree with the Reviewer’s comment. The x- and y-axes now are assigned separately to each figure.
Point 4: Why the combination is working better than individual drugs.
Response 4: As investigation of the mechanism of interactions of daptomycin and gentamicin was not the aim of current research, we did not focus on this question. However, assuming existing knowledge (Ref. #18), we can hypothesize that the possible mechanism responsible for the synergistic interaction between daptomycin (as a cell membrane targeting antibacterial) and gentamicin (ribosomal) is that synergism results if one drug increases the permeability of the cell envelope to the other drug. As referenced in the manuscript, such an uptake effect is likely responsible for the synergism between aminoglycoside and beta-lactam antibiotics. A similar mechanism probably underlies the interaction between daptomycin and gentamicin. We have now included this in the introduction (Lines 50-53). The respective references are included in the reference list (Ref. #18).
Point 5: More detailed conclusions are need to be given.
Response 5: We believe the conclusions are quite inclusive and prefer to leave them unchanged to avoid additional verbiage that would disrupt the general logic of the data presentation.
Point 6: Figure 2 x-axis very difficult to see, modify the figure.
Response 6: We agree with Reviewer’s comment. Now the x-axis labels are increased in size for clarity and emphasis.
Reviewer 2 Report
This study by Golikova et al. reported the prediction of anti-staphylococcal effects of daptomycin and gentamicin combination in an in vitro model and concluded that ∑AUC24/MIC is a reliable predictor. The topic and data in this paper could be of broad interest to readers in the field of antibiotics. Overall, the presented data is of high quality and interpretations drawn are sound. The only comment I have is that most figures in this manuscript seem to lack statistic information. Please take this into account and add such information.
Author Response
Response to Reviewer 2 Comments
Point 1: The only comment I have is that most figures in this manuscript seem to lack statistic information. Please take this into account and add such information.
Response 1: We agree with Reviewer’s comment. The statistical information (standard deviation bars at each data point) for time-kill curves depicted in figure 1 has been inserted.
Reviewer 3 Report
Using AUC/MIC as a metric for antibacterial effects and synergistic effects is not a novel approach. Synergy between daptomycin and gentamicin has been previously studied extensively and abundant literature pre-exists. In my opinion, the manuscript lacks in novelty and significance and therefore it is highly unlikely to attract readers in the field.
Author Response
Response to Reviewer 3 Comments
Obviously, comments provided by Reviewer #3 are general in nature, and they do not require any specific revision of the article. However, we will consider comments to clarify all the points of misconception.
Point 1: Using AUC/MIC as a metric for antibacterial effects and synergistic effects is not a novel approach.
Response 1: This is a misunderstanding. It is possible the Reviewer is confused between the AUC/MIC-based approach to predict antibacterial effects of individual drugs (which is indeed not novel as it has been validated in numerous studies beginning in the 1990s) and our novel approach to predict antibacterial effects of antibiotic combinations using MICs (AUC/MICs) determined at pharmacokinetically-derived concentration ratios as in the current study. The latter approach has not been used by others but was pioneered with initial studies from our laboratory, which introduced this novel approach (Ref. #17, 30). Its applicability was confirmed with other antibiotic combinations, which included either daptomycin or gentamicin. As reflected in the Introduction, the stated aim of the current research was to verify this approach with daptomycin plus gentamicin combinations, (Lines 49-50, 64-67).
In addition, the Reviewer’s sentence “Using AUC/MIC as a metric for…synergistic effects…” is incorrect in another way, as AUC/MIC has not been used as a metric of synergistic effects. To measure the synergy between drugs, special indices and principles exist such as the FIC index and the Bliss independence principle (used in current research), Loewe additivity principle, combination index (CI) and theorem of Chou-Talalay, etc.
Point 2: Synergy between daptomycin and gentamicin has been previously studied extensively and abundant literature pre-exists.
Response 2: Of note, the manuscript includes 13 references (Ref. #19-27, 31-34) to original studies aimed at investigating the efficacy of daptomycin and gentamicin combinations in vitro and/or in vivo and also a review of papers that present exhaustive information in this field. Unfortunately, the papers that actually investigate this antibiotic combination are not as abundant as one might think. In the Introduction section (Lines 53-63) we critically considered the results of experiments with daptomycin-gentamicin combinations presented in existing publications. These results are objectively controversial and reveal an entire spectrum of types of interaction between daptomycin and gentamicin, i.e. synergism/enhancement (Ref. #19, 22, 23, 24-26), additivity or indifference/improvement (Ref. #20, 21, 23, 24-26) and antagonism (Ref. #27). This is clearly stated in the text (Lines 53-63, 177-203). Based on such discrepant data it is difficult to say, “Synergy between daptomycin and gentamicin has been previously studied extensively…” and it is obvious that additional studies are needed to clarify this situation. This sentence was reflected in introduction section and substantiates the relevance of this study (Lines 53-55). Moreover, in the Discussion section the possible reasons for disagreement in different studies with daptomycin plus gentamicin combinations are discussed in detail.
Point 3: In my opinion, the manuscript lacks in novelty and significance and therefore it is highly unlikely to attract readers in the field.
Response 3: As the current study aims to verify with daptomycin/gentamicin our previously published novel approach to predicting the efficacy of other antibacterial combinations, the “novelty and significance” of this study is clearly stated and should be obvious.
Round 2
Reviewer 3 Report
No further comments